# Serological Investigations on West Nile Virus in Horses in Kazakhstan

**DOI:** 10.3390/microorganisms13112541

**Published:** 2025-11-06

**Authors:** Dana A. Alibekova, Kainar B. Barakbayev, Zamira D. Omarova, Rashida A. Rystaeva, Kulyaisan T. Sultankulova, Yerbol D. Burashev, Takhmina U. Argimbayeva, Ali B. Tulendibayev, Nurdos A. Aubakir, Tangat T. Yermekbay, Khairulla B. Abeuov, Aslan A. Kerimbayev, Mukhit B. Orynbayev

**Affiliations:** 1Research Institute for Biological Safety Problems, National Holding QazBioPharm, The Ministry of Health Care of the Republic of Kazakhstan, Gvardeiskiy 080409, Kazakhstanyerbol.bur@gmail.com (Y.D.B.);; 2Research and Production Center “MVA Group”, Village Koksay, Karasai District, Almaty 040921, Kazakhstan

**Keywords:** West Nile virus, ELISA, seroprevalence, horses, Kazakhstan, epidemiology

## Abstract

This study presents the first investigation of West Nile virus (WNV) seroprevalence among farmed horses in Kazakhstan. In 2024, a total of 368 serum samples were collected from horses across 106 settlements in 10 regions of the Republic of Kazakhstan. Using an enzyme-linked immunosorbent assay (ELISA), antibodies to WNV were detected in 32 horses (8.7%; 95% CI: 6.2–12.0%) from six regions. Among the seropositive animals, 26 (81.25%) were females and 6 (18.75%) were males, ranging in age from 1 to 19 years. No statistically significant association between sex and the presence of antibodies to WNV was found in any of the six regions. Significant differences between age groups were observed in Aktobe (χ^2^ = 12.16; *p* = 0.002) and Turkestan (χ^2^ = 4.20; *p* = 0.040). In the remaining regions (Almaty, Zhetisu, West Kazakhstan, and Atyrau), no significant age-related differences were recorded (*p* > 0.05). These findings confirm the circulation of WNV among horse populations in Kazakhstan and highlight the practical value and effectiveness of using horses as sentinel indicators for WNV surveillance.

## 1. Introduction

West Nile virus (WNV) is an arbovirus belonging to the genus *Flavivirus* (family *Flaviviridae*) [1]. The virus is maintained in nature through an enzootic transmission cycle in which ornithophilic mosquitoes—particularly those of the genus *Culex*—act as vectors, while birds serve as amplifying hosts [2]. Recent studies show that environmental factors such as climate change and land-use patterns strongly influence the distribution and abundance of these mosquitoes, and therefore the dynamics of WNV transmission [3,4,5,6]. Humans, horses, and most other mammals are incidental “dead-end” hosts, unable to develop a viremia high enough to infect mosquitoes [7].

Kazakhstan presents a unique ecological landscape that may facilitate WNV transmission. The country’s diverse natural habitats support numerous mosquito species and populations of migratory birds, both key components in the virus’s transmission dynamics [8,9]. Urban areas, in particular, pose elevated risks of WNV transmission due to higher mosquito densities near human settlements [10]. Although humans and horses are considered incidental hosts, they can experience severe clinical manifestations of WNV infection, including encephalitis and other neurological disorders [11].

The first serological evidence of WNV infection in Kazakhstan was reported in 2019, when high titers of anti-WNV IgG were detected in two patients with neuroinvasive symptoms in the Tekeli district of Almaty Region [12]. Since then, concerns have grown regarding WNV spread in Kazakhstan, especially given the detection of the virus in local mosquito populations and the presence of migratory birds that can facilitate its transmission. Despite these findings, data on WNV infection in horses remain scarce, and no official reports of equine cases have been published.

Investigating the seroprevalence of antibodies to WNV in horses is important because horses serve as sentinel indicators of viral circulation and can provide insight into the risk of transmission to humans [13]. The objective of this study is to address this knowledge gap by examining the seroprevalence of WNV among horses in Kazakhstan in 2024, with a focus on regional differences in exposure. The results will enhance understanding of WNV epidemiology in Kazakhstan and inform public-health strategies for monitoring and controlling the virus.

## 2. Materials and Methods

### 2.1. Ethical Statement

All procedures were carried out in accordance with protocols approved by the Institutional Animal Care and Use Committee and in compliance with applicable laws and guidelines (Protocol No. 1_07/14/2023). Animal samples were collected as part of ongoing health surveillance and under the authorization of the Committee for Veterinary Control and Supervision of the Ministry of Agriculture of the Republic of Kazakhstan.

### 2.2. Study Area, Sample Collection, Storage, and Transportation of Biological Specimens

The territory of Kazakhstan covers the area from the eastern outskirts of the Volga Delta in the west to the Altai Mountains in the east, from the West Siberian Plain in the north to the Tien Shan mountain range in the south of the country. The relief of Kazakhstan is mainly flat, with dry steppes covering more than 80% of the country.

The explored territory of Kazakhstan can be divided into several zoogeographical zones. These include steppes and forest-steppes (Akmola region), deserts and semi-deserts (Atyrau, West Kazakhstan, Aktobe, Kyzylorda, and Almaty regions), and mountainous areas (East Kazakhstan and Zhetysu regions).

The climate in eastern Kazakhstan is sharply continental, with hot, moderately dry summers and cold, snowy winters. Western Kazakhstan is characterized by a sharply continental climate with large temperature fluctuations throughout the year and during the day. In the south, the climate is continental with moderately warm winters and hot, long summers.

The diversity of the country’s landscape, climate, and wildlife creates conditions for the existence of various pathogens, primarily associated with ticks and blood-sucking insects.

In 2024, within the framework of the national animal disease monitoring program conducted at farms, a total of 368 horse serum samples were collected across 10 regions of the Republic of Kazakhstan (Table 1, Figure 1). Farm inspections and sample collection were carried out jointly with veterinary officers as part of routine epidemiological surveillance. Prior to sampling, all animals were examined by a veterinarian for clinical signs of disease.

Serum samples were obtained from horses in 106 randomly selected settlements of the Akmola, Almaty, Zhetisu, Turkestan, Abay, East Kazakhstan, West Kazakhstan, Atyrau, Mangystau, and Aktobe regions. All samples were taken from animals showing no clinical signs of any illness. Blood was drawn from the jugular vein using Vacutainer tubes for serum collection (Becton Dickinson, Franklin Lakes, NJ, USA).

Samples were transported to the laboratory in Dewar flasks under controlled conditions at −196 °C and subsequently stored at −70 °C until further analysis. Data recorded for each animal included date of sampling, farm name, village, district, region, age, sex, and the latitude and longitude of the sampling site.

### 2.3. Serological Testing

Serum samples were tested for antibodies against the anti-pr-E antigen of WNV using a commercial ELISA kit (multi-species anti-pr-E antibodies; ID Screen^®^ Flavivirus Competition, Innovative Diagnostics VET, Grabels, France). Serological testing was performed according to the manufacturer’s instructions. Optical density (OD) values were recorded at a wavelength of 450 nm, and results were interpreted following the manufacturer’s specifications.

### 2.4. Western Blotting

Serum samples that tested positive for antibodies to WNV by ELISA were further analyzed using Western blotting. Several studies have demonstrated the usefulness of Western blot analysis for confirming the specificity of other serological tests. Cabré et al. reported high specificity of the Western blot assay, showing 100% correlation between WB and PRNT results [14]. West Nile virus antigen and horse sera diluted 1:100 were used for the assay. Viral proteins were separated by electrophoresis in a 10% polyacrylamide gel under reducing conditions. Proteins were transferred from the gel to a nitrocellulose membrane using a semi-dry blotter (Bio-Rad, South Granville NSW, Australia). After transfer, the membrane was blocked for 1 h at room temperature in 5% nonfat dry milk prepared in TBS with 0.05% Tween 20.

The membrane was then incubated with horse serum (1:100 dilution) to detect antibodies against West Nile virus. Following three washes in TBS/Tween 20, rabbit anti-horse antibodies conjugated with alkaline phosphatase (Sigma, St. Louis, MO, USA) were added at a 1:20,000 dilution for 1 h at room temperature. The membrane was washed three times in TBS/Tween 20 and once in 0.1 M Tris buffer (pH 9.6) before adding the BCIP/NBT liquid substrate system (Sigma) for color development.

### 2.5. Statistical Analysis

Statistical analysis was performed in RStudio v2023.09.1 (Posit Software, PBC, Boston, MA, USA) using R v4.3.2 (R Core Team, Vienna, Austria, 2023). Pearson’s chi-square test (χ^2^) was used to compare seroprevalence across age groups, and Fisher’s exact test was applied to assess differences by sex. Differences were considered statistically significant at *p* < 0.05. Results are presented as percentages with 95% confidence intervals to provide a measure of uncertainty around the estimates. Mapping and spatial visualization were carried out in QGIS 3.26.3 (QGIS Development Team, Küsnacht, Switzerland, 2022).

## 3. Results

Among the 368 horses included in the study, 32 sera tested positive for antibodies to WNV by competitive ELISA (Figure 2). All ELISA-positive sera were subsequently examined by Western blot, and every positive sample was confirmed as specific to West Nile virus (Appendix A).

The overall seroprevalence of antibodies to WNV in the horse population of Kazakhstan was 8.7% (95% CI: 6.2–12.0%). Antibodies were detected in horses from 6 of the 10 surveyed regions (Figure 2 and Figure 3), with seropositive animals identified in 14 settlements. The highest seroprevalence was observed in the Almaty (20.0%, 95% CI: 11.6–32.4%) and the Turkestan (20.0%, 95% CI: 8.1–41.6%), while the lowest rate was recorded in the Atyrau (2.13%, 95% CI: 0.4–10.9%).

Antibodies to WNV were detected in 26 females (81.25%) and 6 males (18.75%) (Table 2). Seropositive horses comprised 20 animals aged 1–5 years, 13 aged 5–10 years, and 2 older than 10 years. Notably, in one farm in Charyn, Uyghur district, Almaty region, four seropositive horses were only one year old, indicating that the most recent WNV infection likely occurred in 2023 or later.

Across the six affected regions, no significant association was found between sex and the presence of antibodies to WNV, with Fisher’s exact test *p*-values ranging from 0.18 (Zhetisu) to 1.00 (Aktobe, Almaty, Atyrau). Significant differences among age groups were observed in Aktobe (χ^2^ = 12.16; *p* = 0.002) and Turkestan (χ^2^ = 4.20; *p* = 0.040). In Aktobe, seroprevalence increased from 0% in horses ≤5 years, to 50% in the 5–10 year group, and 100% in horses ≥10 years. In Turkestan, seroprevalence reached 100% in horses aged 5–10 years, compared with 11% in those ≤5 years. No significant age-related differences were detected in Almaty, Zhetisu, West Kazakhstan, or Atyrau (*p* > 0.05).

## 4. Discussion

In nature, West WNV is maintained through a transmission cycle between mosquito vectors and avian reservoir hosts, while humans and horses act as incidental, dead-end hosts in this cycle [15,16]. Consequently, WNV surveillance in many countries has focused primarily on wild bird populations. Recent studies, however, indicate that horses may serve as more sensitive indicators than birds for the early detection of diseases that pose a threat to humans [17].

This study is the first to investigate the role of horses in the circulation of WNV in Kazakhstan. To assess the impact of WNV in equine populations, 368 serum samples were collected across 10 regions of the country. Unfortunately, the sample size for this study was uneven and small in certain regions. This was due to the fact that horses are kept on pastures year-round throughout almost the entire territory of Kazakhstan. In only some farms, a limited number of animals are housed in stables for fattening for meat production or mare’s milk collection. Therefore, it is very difficult to gain access to animals kept on pastures with free grazing. Additionally, most private farms lack the equipment necessary for restraining large animals (horses) during sample collection. Moreover, horses kept on free-range pastures are semi-wild and undomesticated. Consequently, serum samples were collected only from animals that were available at the time and housed in stables. As noted above, this is the first study conducted in Kazakhstan to determine the circulation of West Nile virus (WNV) in horse populations. Despite the small sample size, the results provide a basis for drawing preliminary conclusions about the prevalence of WNV among horse populations and emphasize the need for continuous monitoring studies across the entire territory of Kazakhstan.

Serological testing revealed antibodies to WNV in 32 horses (8.7%) from six regions, demonstrating that the virus is circulating in these areas. This indicator is lower than in neighboring Russia. The level of seroprevalence among horses in the Saratov region, which borders Kazakhstan, was (15.9 ± 1.1)% [18] In the Volgograd region, it reached 28.0% [19]. The low prevalence level obtained in our study may be associated with the small number of serum samples collected from horses. To accurately determine the prevalence of WNV in the regions, continuous monitoring studies involving a larger number of animals are necessary.

In our studies, antibodies to West Nile virus (WNV) were detected by ELISA and confirmed by Western blot analysis. We did not have the opportunity to perform neutralization tests, which is among the most specific serological method, due to the absence of WNV strains in Kazakhstan’s collections. Available literature reports indicate the use of Western blot as a reliable method to confirm the specificity of other serological tests [14]. Furthermore, it is known that the Envelope (E) protein is the major structural glycoprotein of WNV and plays a key role in binding to cellular receptors and viral entry into the cell [20]. It is considered the main immunogenic component of the viral envelope, inducing the production of neutralizing antibodies. Therefore, detection of antibodies directed against the E protein serves as a reliable confirmation of contact between the animals and the virus and reflects their immune response. Based on these data, Western blot analysis was used in our study to confirm serological specificity.

It should be noted that previously in Kazakhstan, government authorities did not conduct monitoring of WNV. As a result, many equine cases went undetected. The results of these studies will enable evidence-based recommendations for state veterinary authorities to conduct comprehensive investigations of WNV prevalence in horse populations using multiple diagnostic tools (ELISA, neutralization assay, Western blot).

Detection of specific antibodies to WNV in horses indicates that the virus is circulating in various regions of Kazakhstan. In our study, antibodies to WNV were detected among horses in Turkestan (20%), Almaty (20%), Zhetisu (16.36%), Aktobe (15%), West Kazakhstan (6.8%), and Atyrau (2.13%) regions. Apparently, in the areas where seropositive animals were identified, the conditions for the spread and circulation of the virus are more favorable than in other regions. The importance of migratory birds as carriers and hosts of the virus between different territories has been previously reported. Seropositive animals were found in locations situated along migratory routes of wild birds and in wetland habitats favorable for active virus circulation. The territories of the Aktobe, West Kazakhstan, and Atyrau regions provide potential stopover sites for birds migrating from Russia through Kazakhstan toward Europe or Africa. Migration routes also pass through the Middle Eastern countries [21]. Meanwhile, the Turkestan, Almaty, and Zhetisu regions lie along the migratory pathways of birds arriving from Indo-Pakistani wintering grounds, suggesting that the infection might have been introduced by birds coming from WNV-endemic areas. Moreover, our earlier studies confirm that wild migratory birds are carriers of WNV in Kazakhstan. WNV RNA was detected in the southern and southeastern regions of the country in hooded crows (*Corvus corone*), jackdaws (*Corvus monedula*), Eurasian sparrowhawks (*Accipiter nisus*), chiffchaffs (*Phylloscopus collybita*), Turkestan shrikes (*Lanius phoenicuroides*), mallards (*Anas platyrhynchos*), great cormorants (*Phalacrocorax carbo*), common whitethroats (*Sylvia communis*), common sandpipers (*Actitis hypoleucos*), and Caspian gulls (*Larus cachinnans*). PCR-positive birds were identified along the Aksu River and at the Sorbulak and Alakol lakes [22,23,24]. Most likely, they play a significant role in maintaining and sustaining WNV foci. Occupying quite diverse ecological niches, these bird species may be natural reservoirs of the WNV.

Observed regional differences in seroprevalence may be driven by multiple factors, including environmental conditions, vector distribution, and migratory bird pathways. For example, the higher seroprevalence detected in the Aktobe and Turkestan regions likely reflects ecological settings favorable for mosquito proliferation and increased contact between horses and infected vectors. These findings highlight a potential risk of WNV transmission not only to equines but also to humans.

Our data indicate that WNV circulates among horses in Kazakhstan independently of sex, consistent with previous studies reporting no association between horse sex and risk of WNV infection [25]. Age emerged as a significant risk factor only in certain regions (Aktobe and Turkestan), where seroprevalence increased markedly in older animals, reflecting the cumulative nature of exposure and underscoring the need to consider age in surveillance and vaccination strategies. The significant differences observed in these regions show a clear trend: the older the animal, the higher the likelihood of detecting antibodies. This pattern is typical for infections with natural circulation, where the probability of exposure rises over time. High seroprevalence among horses aged 5–10 years and especially those ≥10 years may reflect long-term contact with vectors such as mosquitoes and ticks and the gradual accumulation of an immune response. The absence of age-related differences in other regions may be related to lower intensity of viral circulation or to seasonal vector activity. Aktobe (a semi-desert zone) and Turkestan regions are characterized by warm climates and prolonged vector activity, which could facilitate greater cumulative exposure of animals. Small sample sizes in specific age subgroups (particularly ≥10 years) may also have limited the power to detect differences. Further studies with larger age-stratified cohorts are needed to clarify the effect of age.

Despite the absence of reported clinical WNV cases in horses in Kazakhstan, the observed seroprevalence underscores the value of horses as sentinel indicators of WNV activity. Horses are highly susceptible to severe manifestations of WNV infection, making them an important component for assessing potential human risk. Our findings align with previous research identifying equines as critical elements in WNV epidemiology and highlight the need for targeted surveillance and vaccination strategies in high-risk areas [13,26].

Moreover, the lack of comprehensive surveillance and vaccination programs in some regions of Kazakhstan raises concerns about the vulnerability of equine populations to WNV infection. Implementing proactive surveillance measures and vaccination campaigns in identified hotspots could reduce transmission risk and protect both horses and humans.

## 5. Conclusions

In conclusion, this study demonstrates that WNV circulates among horses in regions of Kazakhstan and emphasizes the need for ongoing research and surveillance efforts to monitor WNV in horse populations within the country. Our findings have increased awareness of the prevalence of WNV among horses in Kazakhstan. However, the available data are insufficient to fully understand the risk of WNV spread across various ecological zones. Further studies on natural reservoirs and vectors of WNV in different regions are necessary to assess the current epidemiological situation in the country. Understanding the dynamics of WNV transmission is essential for developing effective public health strategies and safeguarding the health of both horses and people. Future research should focus on the interactions among environmental factors, vector populations, and host susceptibility to achieve a more comprehensive understanding of WNV epidemiology in the region.

## Figures and Tables

**Figure 1 microorganisms-13-02541-f001:**
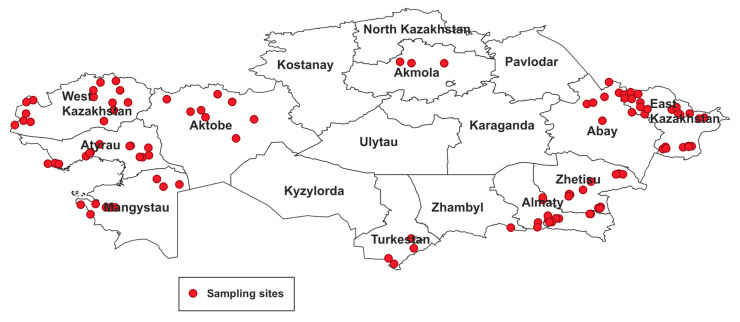
Study area and sample collection sites.

**Figure 2 microorganisms-13-02541-f002:**
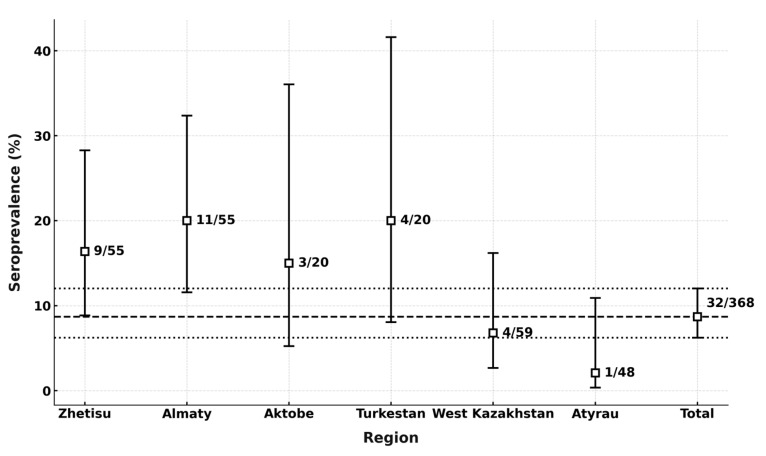
Seroprevalence of WNV by region. The dashed lines represent the mean seroprevalence (8.7%) across all regions (central line) and the 95% confidence interval (upper and lower lines: 6.2–12.0%). These indicators reflect the overall prevalence of antibodies to WNV in the total sample and the statistical variability of the estimate.

**Figure 3 microorganisms-13-02541-f003:**
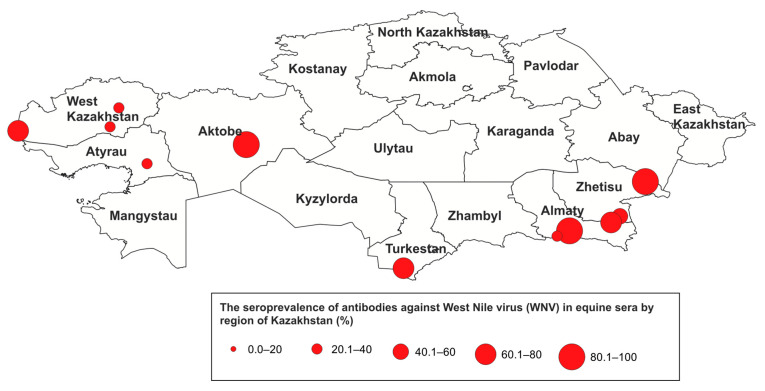
Seroprevalence of WNV in the Republic of Kazakhstan in 2024.

**Table 1 microorganisms-13-02541-t001:** Number of surveyed regions and characteristics of collected samples.

No.	Region	Sampling Date	Number of Tested		Number of Collected Samples			
			Districts	Settlements	Age	Female	Male	Total
1	Akmola	23–29 May 2024	3	3	≤5 years	6	2	12
					5–10 years	4	0	
					≥10 years	0	0	
2	Almaty	30 April–14 May 2024	5	13	≤5 years	42	7	55
					5–10 years	5	0	
					≥10 years	1	0	
3	Zhetisu	23–29 April 2024	4	11	≤5 years	23	8	55
					5–10 years	22	2	
					≥10 years	0	0	
4	Turkestan	2–5 May 2024	4	4	≤5 years	5	13	20
					5–10 years	0	2	
					≥10 years	0	0	
5	Abay	27 May–8 June 2024	4	11	≤5 years	20	2	22
					5–10 years	0	0	
					≥10 years	3	0	
6	East Kazakhstan	17–25 May 2024	6	19	≤5 years	24	4	38
					5–10 years	6	4	
					≥10 years	0	0	
7	West Kazakhstan	12–20 June 2024	6	16	≤5 years	26	9	59
					5–10 years	18	2	
					≥10 years	4	0	
8	Atyrau	18–28 August 2024	4	12	≤5 years	15	8	48
					5–10 years	16	5	
					≥10 years	3	1	
9	Mangystau	8–16 August 2024	3	9	≤5 years	18	2	39
					5–10 years	14	1	
					≥10 years	3	1	
10	Aktobe	1–8 July 2024	6	8	≤5 years	13	2	20
					5–10 years	4	0	
					≥10 years	1	0	
Total			45	106	≤5 years	192	57	368
					5–10 years	89	16	
					≥10 years	12	3	

**Table 2 microorganisms-13-02541-t002:** Seroprevalence by sex and age group across regions.

No.	Region	Characteristics	n	Positives	%	*p*-Value
1	Zhetisu	**Sex**				0.186
		Female	45	9	100%	
		Male	10	0	0%	
		**Age**				1.000
		≤5 years	31	5	16.1%	
		5–10 years	24	4	16.7%	
		≥10 years	0	0	—	
2	Almaty	**Sex**				1.000
		Female	48	10	20.8%	
		Male	7	1	14.3%	
		**Age**				0.880
		≤5 years	49	10	20.4%	
		5–10 years	5	1	20%	
		≥10 years	1	0	0%	
3	Aktobe	**Sex**				1.000
		Female	18	3	16.7%	
		Male	2	0	0%	
		**Age**				0.002
		≤5 years	15	0	0%	
		5–10 years	4	2	50%	
		≥10 years	1	1	100%	
4	Turkestan	**Sex**				0.530
		Female	5	0	0%	
		Male	15	4	26.7%	
		**Age**				0.002
		≤5 years	18	2	11.1%	
		5–10 years	2	2	100%	
		≥10 years	0	0	—	
5	West Kazakhstan	**Sex**				0.572
		Female	48	3	6.2%	
		Male	11	1	9.1%	
		**Age**				0.194
		≤5 years	35	1	5.7%	
		5–10 years	20	2	10%	
		≥10 years	4	1	25%	
6	Atyrau	**Sex**				1.000
		Female	34	1	2.9%	
		Male	14	0	0%	
		**Age**				0.574
		≤5 years	23	1	4.3%	
		5–10 years	21	0	0%	
		≥10 years	4	0	0%	

**Notes:** “χ^2^”—value of the Pearson’s chi-square test, calculated only for age groups. “*p*-value” for sex was obtained using Fisher’s exact test. *p* < 0.05 was considered statistically significant.

## Data Availability

The data presented in this study are available on request from the corresponding author. The data are not publicly available due to privacy.

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
