# Peer review of "Serological Investigations on West Nile Virus in Horses in Kazakhstan"

_microorganisms, 2025, doi:10.3390/microorganisms13112541_

Round 1
Reviewer 1 Report
Comments and Suggestions for Authors
In this study, horse serum samples from 10 regions of the Republic of Kazakhstan were tested for WNV using a commercially available ELISA, with confirmation by western blot analysis. Anti-WNV antibodies were detected in 8.7% of horses sampled across six regions. The work is clearly presented and supported by appropriate statistical analysis. The authors demonstrate a solid understanding of the complexities of WNV and its potential threat to humans. The manuscript is well written and does not require English language editing. However, the reference list should be updated with more relevant and appropriate sources.
Minor comments:
The title needs to be changed. E.g. Serological Investigations on West Nile Virus positive Horses in Kazakhstan (or
Abstract
Please include if these are farmed horses or wild horses
This sentence is not needed: (Fisher’s exact test p-values ranged from 0.18 in Zhetisu to 1.00 in Aktobe, Almaty, and Atyrau).
Manuscript
Line 66: In 2024, within the framework of the national animal disease monitoring program conducted at farms
Line 84 &168: Don’t need to abbreviate WNV again. Just use WNV. Please use WNV abbreviation throughout after the first use of the full name.
Line 117: Elaborate in the figure legend. Explain what the dotted lines (95% CI and average?) are.
It’s difficult to read the names on figure 1 and 3. Please use a different font or make it bigger.
Line 150: There are better papers to reference than reference 16. E.g. https://www.sciencedirect.com/science/article/pii/S0378113507005809
Line 154: Can not say “actively” unless you tested the horses or mosquitoes for viral RNA. ELISA data showed that WNV circulated at some stage. Migratory birds could bring it in seasonally. (line 158)
Line 155-156: Please elaborate. Can the authors discuss the environmental conditions between the areas, maybe in the methods list e.g grassland or altitude. It would be interesting to see if altitude could explain the negative population (higher = colder weather), or maybe the lack of urban areas around the farm could explain less C.pipiens in the area.
Author Response
Dear reviewer!
Thank you for your positive response to our article that was recently submitted to Microorganisms.
Below is a detailed response to the suggestions.
Minor comments:
The title needs to be changed. E.g. Serological Investigations on West Nile Virus positive Horses in Kazakhstan (or
Response: We thank the reviewer for the valuable comment.
After careful consideration, we decided to keep the original title, as it best represents the study’s objectives and key findings.
Comment 1: Abstract. Please include if these are farmed horses or wild horses
This sentence is not needed: (Fisher’s exact test p-values ranged from 0.18 in Zhetisu to 1.00 in Aktobe, Almaty, and Atyrau).
Response: It has been done. We included in the abstract that the horses are farm horses. At your recommendation, we deleted the sentence: (Fisher’s exact test p-values ranged from 0.18 in Zhetisu to 1.00 in Aktobe, Almaty, and Atyrau).
Comment 2: Manuscript. Line 66: In 2024, within the framework of the national animal disease monitoring program conducted at farms
Response: It has been done. We have changed the sentence as you suggested..
Comment 3: Line 84 &168: Don’t need to abbreviate WNV again. Just use WNV. Please use WNV abbreviation throughout after the first use of the full name.
Response: It has been done.
Comment 4: Line 117: Elaborate in the figure legend. Explain what the dotted lines (95% CI and average?) are.
Response: It has been done. Explanations for the figure are included.
Comment 5: It’s difficult to read the names on figure 1 and 3. Please use a different font or make it bigger.
Response: It has been done. The figures have been changed.
Comment 6: There are better papers to reference than reference 16. E.g. https://www.sciencedirect.com/science/article/pii/S0378113507005809
Response: It has been done. The article has been replaced.
Comment 7: Line 154: Can not say “actively” unless you tested the horses or mosquitoes for viral RNA. ELISA data showed that WNV circulated at some stage. Migratory birds could bring it in seasonally. (line 158)
Response: It has been done.
Comment 8: Line 155-156: Please elaborate. Can the authors discuss the environmental conditions between the areas, maybe in the methods list e.g grassland or altitude. It would be interesting to see if altitude could explain the negative population (higher = colder weather), or maybe the lack of urban areas around the farm could explain less C.pipiens in the area.
Response: It has been done.
Reviewer 2 Report
Comments and Suggestions for Authors
This study is the first to investigate the serum prevalence of West Nile virus (WNV)
Antibody levels in horses in Kazakhstan. These findings confirm the prevalence of West Nile virus in horse herds in Kazakhstan and its practical value and benefits the effectiveness of using horses as sentinel indicators for monitoring West Nile virus. However, the main issues with the article are as follows:
- The article design is too simple and lacks any molecular marker discovery and validation experiments, making it difficult to convince people.
- 368 serum samples were collected from horses in 106 settlements across 10 regions of the Republic of Kazakhstan. Using enzyme-linked immunosorbent assay (ELISA) to detect positive yellow virus antibodies, is the distribution of this sample comprehensive, reasonable, and representative? Additionally, relying solely on ELISA for detection is very limited, and two or three methods should be used to simultaneously validate this result.
- These findings confirm the prevalence of West Nile virus in horse herds in Kazakhstan, but the conclusions of the article are far from being drawn from existing simple data, and their practical significance and value are also very limited.I really can't continue commenting.
Author Response
Dear reviewer!
Thank you for your positive response to our article that was recently submitted to Microorganisms.
Below is a detailed response to the suggestions.
- The article design is too simple and lacks any molecular marker discovery and validation experiments, making it difficult to convince people.
- 368 serum samples were collected from horses in 106 settlements across 10 regions of the Republic of Kazakhstan. Using enzyme-linked immunosorbent assay (ELISA) to detect positive yellow virus antibodies, is the distribution of this sample comprehensive, reasonable, and representative? Additionally, relying solely on ELISA for detection is very limited, and two or three methods should be used to simultaneously validate this result.
- These findings confirm the prevalence of West Nile virus in horse herds in Kazakhstan, but the conclusions of the article are far from being drawn from existing simple data, and their practical significance and value are also very limited. I really can't continue commenting.
Comment 1: The article design is too simple and lacks any molecular marker discovery and validation experiments, making it difficult to convince people.
Response: To confirm the presence of antibodies against West Nile virus (WNV), a Western blot analysis was performed using the viral envelope antigen. The E protein is the major structural glycoprotein of WNV and plays a key role in receptor binding and viral entry into host cells [Kuhn RJ, Zhang W, Rossmann MG, Pletnev SV, Corver J, Lenches E, Jones CT, Mukhopadhyay S, Chipman PR, Strauss EG, Baker TS, Strauss JH. Structure of dengue virus: implications for flavivirus organization, maturation, and fusion. Cell 2002;108:717–725]. It is considered the principal immunogenic component of the viral envelope that induces the production of neutralizing antibodies. Therefore, the detection of antibodies directed against the E protein serves as reliable evidence of animal exposure to the virus and reflects the immune response.
Several studies have demonstrated the usefulness of Western blot analysis for confirming the specificity of other serological tests. For instance, Cabré et al. reported high specificity of the Western blot assay, showing 100% correlation between WB and PRNT results on a panel of 79 horse serum samples [Cabré, O.; Grandadam, M.; Marié, J.-L.; Gravier, P.; Prangé, A.; Santinelli, Y.; Rous, V.; Bourry, O.; Durand, J.-P.; Tolou, H.; Davoust, B. West Nile Virus in Horses, sub-Saharan Africa. Emerg. Infect. Dis. 2006, 12(12), 1958–1960. https://doi.org/10.3201/eid1212.060042].
Comment 2: 368 serum samples were collected from horses in 106 settlements across 10 regions of the Republic of Kazakhstan. Using enzyme-linked immunosorbent assay (ELISA) to detect positive yellow virus antibodies, is the distribution of this sample comprehensive, reasonable, and representative? Additionally, relying solely on ELISA for detection is very limited, and two or three methods should be used to simultaneously validate this result.
Response: We agree with the reviewer that the sample size in some regions was relatively small. When planning the study, we aimed to cover as many regions of Kazakhstan as possible. However, during field expeditions, we were unable to collect the initially planned number of animal samples. This limitation was primarily due to the fact that, throughout most of Kazakhstan, horses are kept on pastures year-round. Only in a few farms are small numbers of animals kept in enclosures either for meat production or for obtaining mare’s milk.
Because of this management system, it is very difficult to access horses that are kept on open pastures. In addition, most private farms are not equipped with facilities for restraining large animals such as horses. Horses that are freely grazing are typically untrained or semi-feral, which makes blood sampling extremely challenging. Therefore, serum samples were collected only from animals that were available at the time and kept under stable (stall) conditions.
This study represents the first investigation in Kazakhstan aimed at determining the circulation of West Nile virus in horse populations. Despite the limited sample size in some regions, the results provide valuable preliminary evidence of WNV exposure in horses and emphasize the need for large-scale surveillance studies to further clarify these findings.
Comment 3: These findings confirm the prevalence of West Nile virus in horse herds in Kazakhstan, but the conclusions of the article are far from being drawn from existing simple data, and their practical significance and value are also very limited.I really can't continue commenting.
Response: We agree with the reviewer that reliable serological studies should ideally employ two or three complementary diagnostic methods. The virus neutralization test (VNT) is indeed considered the gold standard in serological investigations. However, due to the absence of West Nile virus (WNV) in our institute’s virus collection, we were unable to perform this assay. Currently, WNV is not available in any reference collection in Kazakhstan.
As mentioned above, there is precedent for the use of Western blot analysis to confirm ELISA results. Therefore, in our study, Western blot was applied as a secondary method to verify the ELISA findings.
It is important to note that this study provides the first serological evidence of WNV circulation among horses in Kazakhstan. Previously, no official monitoring of WNV infection had been conducted by veterinary authorities, and many potential cases among horses likely went unnoticed. This research thus represents the first pilot study on WNV seroprevalence in horses in Kazakhstan.
The obtained results provide a foundation for evidence-based recommendations to national veterinary authorities to establish continuous surveillance programs. Based on our findings, we have developed preliminary recommendations for future monitoring of WNV in horse populations, as well as in other livestock species across the territory of Kazakhstan.
Reviewer 3 Report
Comments and Suggestions for Authors
There are minor concerns that need to be addressed to make the manuscript better.

Author Response
Dear reviewer!
Thank you for your positive response to our article that was recently submitted to Microorganisms.
Below is a detailed response to the suggestions.
Comment:
This manuscript is relevant since it addresses a global concern of emerging mosquitoborne viruses and environmental factors favoring their spread in Kazakhstan. It addresses a critical gap in knowledge by being the 1st group to assess the prevalence of West Nile Virus (WNV) exposure in horses through serological testing. However, there are some concerns which need to be addressed to make the manuscript stronger.
Major concerns:
Comment 1: A major drawback of the study is varying sample sizes from different regions like Akmola, Turkestan and Aktobe have the smallest sample size compared to all other regions. This makes the interpretations, based on the statistical analyses, unreliable.
Response: We agree with the reviewer that the sample size in some regions was relatively small. When planning the study, we aimed to cover as many regions of Kazakhstan as possible. However, during field expeditions, we were unable to collect the initially planned number of animal samples. This limitation was primarily due to the fact that, throughout most of Kazakhstan, horses are kept on pastures year-round. Only in a few farms are small numbers of animals kept in enclosures either for meat production or for obtaining mare’s milk.
Because of this management system, it is very difficult to access horses that are kept on open pastures. In addition, most private farms are not equipped with facilities for restraining large animals such as horses. Horses that are freely grazing are typically untrained or semi-feral, which makes blood sampling extremely challenging. Therefore, serum samples were collected only from animals that were available at the time and kept under stable (stall) conditions.
This study represents the first investigation in Kazakhstan aimed at determining the circulation of West Nile virus in horse populations. Despite the limited sample size in some regions, the results provide valuable preliminary evidence of WNV exposure in horses and emphasize the need for large-scale surveillance studies to further clarify these findings.
Comment 2: The authors mention the use of a multi-species Flavivirus ELISA and although these were confirmed by Western blots, authors need to mention the possibility of crossreactivity with other flavivirus antibodies and how did they ensure that the antibodies were specific to WNV.
Response: To confirm the presence of antibodies against West Nile virus (WNV), a Western blot analysis was performed using the viral envelope antigen. The E protein is the major structural glycoprotein of WNV and plays a key role in receptor binding and viral entry into host cells [Kuhn RJ, Zhang W, Rossmann MG, Pletnev SV, Corver J, Lenches E, Jones CT, Mukhopadhyay S, Chipman PR, Strauss EG, Baker TS, Strauss JH. Structure of dengue virus: implications for flavivirus organization, maturation, and fusion. Cell 2002;108:717–725]. It is considered the principal immunogenic component of the viral envelope that induces the production of neutralizing antibodies. Therefore, the detection of antibodies directed against the E protein serves as reliable evidence of animal exposure to the virus and reflects the immune response.
Several studies have demonstrated the usefulness of Western blot analysis for confirming the specificity of other serological tests. Cabré et al. reported high specificity of the Western blot assay, showing 100% correlation between WB and PRNT results on a panel of 79 horse serum samples [Cabré, O.; Grandadam, M.; Marié, J.-L.; Gravier, P.; Prangé, A.; Santinelli, Y.; Rous, V.; Bourry, O.; Durand, J.-P.; Tolou, H.; Davoust, B. West Nile Virus in Horses, sub-Saharan Africa. Emerg. Infect. Dis. 2006, 12(12), 1958–1960. https://doi.org/10.3201/eid1212.060042].
Comment 3: The authors establish the importance of mosquitoes and birds in the context of WNV transmission, but there is no discussion or mention of the mosquitoes or birds in Kazakhstan, which is another major limitation of this study.
Response: Thank you for the valuable comment. Indeed, mosquitoes play an important role in the transmission of this disease. Previously, Nurmakhanov et al. detected WNV RNA in Culex modestus mosquitoes in the West Kazakhstan region [Nurmakhanov T., Sansyzbaev Y., Atshabar B., Berlin V., Kobzhasarov D., Yeskhojayev O., Vilkova A., Ayazbayev T., Andryuchshenko A., Bidashko F., Hay J., and Shvetsov A. (2021). Phylogenetic Characteristics of West Nile Virus Isolated From Culex modestus Mosquitoes in West Kazakhstan. Front. Public Health 8:575187. https://doi.org/10.3389/fpubh.2020.575187].
We have added this information to the Discussion section of the manuscript. Our study, however, was focused specifically on the serological investigation of horses, which represent one of the key sentinel species for monitoring the circulation of WNV.
Comment 4: How do the results from this study compare with areas with prevalent WNV cases, in terms of seroprevalence, vectors and vertebrate host susceptibility?
Response: In the revised version of the manuscript, we have added data on the seroprevalence of West Nile virus among horses in the border regions of Russia to provide a broader regional context for our findings.
Reviewer 4 Report
Comments and Suggestions for Authors
This study presents the first nationwide investigation of West Nile virus (WNV) seroprevalence among horses in Kazakhstan, aiming to assess the extent of WNV exposure and its regional distribution. By analyzing 368 equine serum samples collected from 10 regions using ELISA, the study found an overall seroprevalence of 8.7%, confirming active WNV circulation. Significant age-related differences in seropositivity were observed in two regions, while no association with sex was found. The main contribution of this research lies in its provision of the first serological data on WNV in Kazakhstan equine populations, supporting the utility of horses as sentinel species for WNV surveillance. The study’s strengths include its broad geographic coverage, robust sampling methodology, and relevance to both veterinary and public health monitoring efforts.
The manuscript is well-structured, with a clear organization of sections that logically present the background, methodology, results, and their interpretation. There are only minor issues (see comments below)
Minor issues:
Materials and methods
Line 67: There's a discrepancy; you stated that the samples were collected from 10 regions, whereas the Figure 1 shows 11. Please correct this to be consistent.
Line 91: Could you elucidate for readers the justification for utilizing WB o flavivirus antibody-positive samples.
In statistical analysis (line 104) add manufacturer names for statistical package and visualization software.
Please provide WB gel in supplementary file.
Results
Explain Figure 2 in more detail - what do the dashed horizontal lines represent, you stated fractions instead of percentages (y-axis)
Figure 1 shows the Jetisu area, while Table 1 shows the Zhetisu area, please be uniform.
Discussion
Lines 153-154: You reported a seroprevalence of 8.7% in horses. To enhance the contextual significance of this finding, it would be valuable to compare it with data from neighboring countries or similar epidemiological settings.
Lines 165-168: you stated “likely reflects ecological settings…”. Please be more specific.
Reference
The references used are relevant and appropriate for the research topic. Notably, approximately 50% of the cited sources were published within the last five years (2020–2025), indicating a high level of up-to-date and reliance on current scientific data.
Author Response
Dear reviewer!
Thank you for your positive response to our article that was recently submitted to Microorganisms.
Below is a detailed response to the suggestions.
The manuscript is well-structured, with a clear organization of sections that logically present the background, methodology, results, and their interpretation. There are only minor issues (see comments below)
Minor issues:
Comment: Materials and methods. Line 67: There's a discrepancy; you stated that the samples were collected from 10 regions, whereas the Figure 1 shows 11. Please correct this to be consistent.
Response: It has been corrected.
Comment: Line 91: Could you elucidate for readers the justification for utilizing WB o flavivirus antibody-positive samples.
Response: It has been done.
Comment: Line 91: In statistical analysis (line 104) add manufacturer names for statistical package and visualization software.
Response: It has been done.
Comment: Please provide WB gel in supplementary file.
Response: It has been done. We have added the figure to the supplementary material.
Comment: Results. Explain Figure 2 in more detail - what do the dashed horizontal lines represent, you stated fractions instead of percentages (y-axis)
Response: It has been done.
Comment: Figure 1 shows the Jetisu area, while Table 1 shows the Zhetisu area, please be uniform.
Response: It has been changed.
Comment: Discussion. Lines 153-154: You reported a seroprevalence of 8.7% in horses. To enhance the contextual significance of this finding, it would be valuable to compare it with data from neighboring countries or similar epidemiological settings.
Response: It has been done. We have added data on neighboring countries to the discussion text.
Comment: Lines 165-168: you stated “likely reflects ecological settings…”. Please be more specific.
Response: It has been done.
Round 2
Reviewer 2 Report
Comments and Suggestions for Authors
I believe the quality of the revised manuscript has improved, but I feel my key concerns have not been adequately addressed. I must adhere to my rigorous review standards.
Author Response
In accordance with your recommendations, we have made changes to the text of the article.
To assist readers in correctly interpreting the results, we added further explanations in the article text regarding the difficulties of sample collection. We also included data on the limitations of the neutralization reaction and the rationale for selecting western blot to confirm specificity.
We replaced Figure 2 with a higher-quality version.